# From data strategy to implementation to advance cancer research and cancer care: A French comprehensive cancer center experience

**Pierre Heudel**[1]*, **Hugo Crochet**[2], **Thierry Durand**[3], **Philippe Zrounba**[4], **Jean-Yves Blay**[1,5]

1 Department of Medical Oncology, Centre Léon Bérard, Lyon, France, 2 Data and Artificial Intelligence Team, Centre Léon Bérard, Lyon, France, 3 Data protection officer, Centre Léon Bérard, Lyon, France, 4 Department of Surgical Oncology, Centre Léon Bérard, Lyon, France, 5 General Director, Centre Léon Bérard, Lyon, France

☯ These authors contributed equally to this work.

\* Pierreetienne.heudel@lyon.unicancer.fr

**Data Availability Statement:** In accordance with French regulations controlled by the CNIL

## Abstract

In a comprehensive cancer center, effective data strategies are essential to evaluate practices, and outcome, understanding the disease and prognostic factors, identifying disparities in cancer care, and overall developing better treatments. To achieve these goals, the Center Léon Bérard (CLB) considers various data collection strategies, including electronic medical records (EMRs), clinical trial data, and research projects. Advanced data analysis techniques like natural language processing (NLP) can be used to extract and categorize information from these sources to provide a more complete description of patient data. Data sharing is also crucial for collaboration across comprehensive cancer centers, but it must be done securely and in compliance with regulations like GDPR. To ensure data is shared appropriately, CLB should develop clear data sharing policies and share data in a controlled, standardized format like OSIRIS RWD, OMOP and FHIR. The UNICANCER initiative has launched the CONSORE project to support the development of a structured and standardized repository of patient data to improve cancer research and patient outcomes. Real-world data (RWD) studies are vital in cancer research as they provide a comprehensive and accurate picture of patient outcomes and treatment patterns. By incorporating RWD into data collection, analysis, and sharing strategies, comprehensive cancer centers can take a more comprehensive and patient-centered approach to cancer research. In conclusion, comprehensive cancer centers must take an integrated approach to data collection, analysis, and sharing to enhance their understanding of cancer and improve patient outcomes. Leveraging advanced data analytics techniques and developing effective data sharing policies can help cancer centers effectively harness the power of data to drive progress in cancer research.

(Commission nationale de l'informatique et des libertés), The data that support the findings of this study are not publicly available. Data contain information that could compromise privacy of the research participants. Requests for access can be made to: Commission nationale de l'informatique et des libertés. Tel: +33 (0) 1 53 73 22 22 Mail: presse@cnil.fr https://www.cnil.fr/fr/saisir-la-cnil/contacter-la-cnil-standard-et-permanences-telephoniques.

**Funding:** The authors received no specific funding for this work.

**Competing interests:** The authors have declared that no competing interests exist.

## Author summary

As soon as I arrived as a medical oncologist at the Léon Bérard center, a French comprehensive cancer center, in 2010, it seemed essential to me to develop our electronic medical record in order to help all health professionals in the care of cancer patients but also have the possibility of reusing the health data produced daily. Since the beginning of the 20th century and the arrival of X-rays and radium therapy in the care of cancer patients, the use of real-world data to the evaluation of medical practice has been done gradually in order to appreciate the contribution of these new treatments compared to surgery which was the only possible treatment at the time. Today, the digitization of our health system makes it possible to carry out more complex studies, on a larger scale and above all more quickly. To do this, however, each cancer center must implement a data strategy to facilitate the collection of this data, its analysis and its sharing with the aim of further accelerating cancer research.

## Introduction

Cancer is a disease with multidimensional complexity that affects millions of people worldwide, and it requires a comprehensive approach for effective treatment and management [1,2]. Comprehensive cancer centers play a crucial role in providing state-of-the-art cancer care and conducting cutting-edge research to advance our understanding of the disease [3]. To effectively fulfill these roles, comprehensive cancer centers need to adopt effective data strategies that enable the efficient collection, analysis, and use of data [4,5]. Contending with a continuously expanding volume and a variety of clinical data poses challenges and opportunities for a comprehensive cancer center. For several years, scientific cancer societies have been able to describe their data strategies, but essentially on a national scale and with a very (too?) theoretical approach [6–9]. This theoretical and generalist approach is essential because it must be shared, but its practical application can most often only be developed at the institutional level.

Data from EMR and all databases are a crucial asset in comprehensive cancer centers, as it provides valuable information about patient care, disease progression, and treatment outcomes. This information can be used to inform clinical decision-making, improve patient outcomes, and drive research advancements and even more with the rise of artificial intelligence [10,11]. Furthermore, data can also be used to monitor and evaluate the performance of cancer centers, which is essential for quality improvement initiatives and accreditation [12,13]. It is therefore essential for such hospital structures to define a strategy for collecting and analyzing data, as well as sharing health data in compliance with the regulatory constraints in force.

In this article, we present the global data strategy of a French comprehensive cancer center, providing concrete examples of practical implementation. Our aim is to share the advantages and difficulties of this institutional organization, illustrating how a well-designed data strategy can contribute to improved patient care and research advancement in the complex world of cancer treatment and management.

## Materials and methods

### Centre Léon Bérard

The French comprehensive cancer center Léon Bérard, located in the Rhône-Alpes region, launched in 1989 a pilot phase to implement a single EMR procedure for each cancer patient. This procedure was adopted from January, 1993, and computerized medical observations and

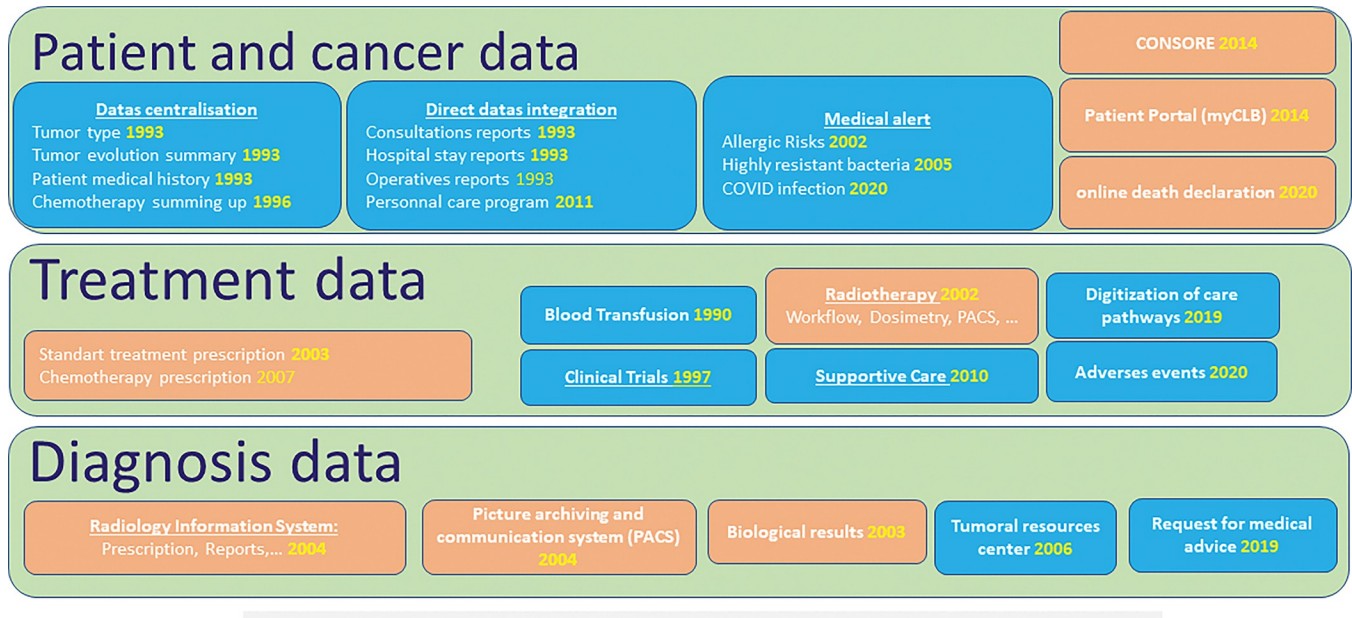

**Fig 1. Patient-centered Applications.** (In yellow, date of current use).

hospitalization reports, then progressively integrated prescriptions of chemotherapy and blood products since 1996, external medical reports since 2000, and additional files since 2005 such as medical imaging, histology results, other medication, etc. In July 2002, EMR became the reference file, and a full electronic format for medical records was definitively adopted in 2006 with regular updates performed on a daily basis. Since 2015, a patient portal « myCLB » has allowed cancer patients to access their medical records, schedule appointments, and report their symptoms on online forms. Computerization of medical records opens up opportunities for profound change in data sources availability on extended medical information on cancer management. For this purpose, the CLB has an information system directorate, which includes a unit managing IT infrastructure and cybersecurity (10 persons), a unit managing internal development of IT solutions and interfaces with business software (10 persons), and since 2021, a data factory (10 persons) aimed at promoting data structuring and facilitating its reuse. The whole is directed by a couple doctor/engineer allowing to have a dual approach and requiring a close collaboration (S1 Fig). Fig 1 represents all the computer software used, specifying their years of installation and whether they were developed in internally by our IT department or purchased from an external service provider. For example, the Biological Resources Center and the information system allow us to specify, for each patient, all quantities of biological materials (tumors, blood, plasma, etc.) that are still available. The medical review form is a simple digital form that nevertheless enables the collection of digital data (dermatology photos, indications for oncogenetic investigations, or specific symptoms for gynecological consultation requests.).

## Tools

*CONSORE.* Since 2015, CLB actively participated in the development and deployment of CONSORE [14]. The CONSORE project is a component of the UNICANCER initiative, a network of comprehensive cancer centers in France. The goal of the CONSORE project is to

develop a decentralized and standardized repository of patient data to support cancer research and improve patient outcomes. CONSORE aims to use data from multiple sources, including electronic medical records (EMRs), pathology reports, and clinical trial data, to create a comprehensive and up-to-date view of patient data. The project uses advanced data analytics and NLP (Natural Language Processing) techniques to extract, structure, and analyze data, with the goal of providing insights into patient outcomes, treatment patterns, and other relevant information. CONSORE represents an important step forward for comprehensive cancer centers in France, as it will enable them to better understand the disease, identify disparities and unmet needs in cancer care, and develop more effective treatments. By leveraging the collective resources of the UNICANCER network, the CONSORE project has the potential to greatly enhance our understanding of cancer and improve patient outcomes.

*OSIRIS*. One approach to standardized data collection and analysis in comprehensive cancer centers is the use of common data models, such as the OSIRIS model [15]. The OSIRIS model is a data model specifically designed for oncology research and provides a standardized framework for collecting and analyzing cancer data. By using a common data model like OSIRIS, comprehensive cancer centers can ensure that their data collection, analysis, and sharing strategies are consistent and comparable. This can facilitate the sharing of data between institutions and researchers and enable the development of more robust and informative analysis techniques. By adopting a common data model, comprehensive cancer centers can take a collaborative approach to cancer research and improve our understanding of this complex disease.

## General Data Protection Regulation (GDPR)

In the European context, data collection, analysis, and sharing strategies in comprehensive cancer centers are governed by the General Data Protection Regulation (GDPR). The GDPR provides a framework for protecting the privacy and security of personal data, including health data. Comprehensive cancer centers must ensure that their data collection, analysis, and sharing strategies comply with the GDPR and protect patient privacy. This may involve obtaining patient consent for the collection and use of personal data, implementing appropriate security measures to protect the confidentiality of patient information, and regularly reviewing and updating data collection and analysis procedures to ensure compliance with the GDPR. Failure to comply with the GDPR can result in significant fines and reputational damage, so it is essential for a French comprehensive cancer centers to take a proactive approach to data privacy and security. To achieve this, we have established two levels of patient information. Firstly, general information is displayed in our premises, sent through our patient portal myCLB, and included in the welcome booklets provided to all new patients [16]. In addition to this general information, individual patient information is included in these retrospective studies. It must be performed for each project in which the patient participates or for which the patient's data will be processed. Thus, each patient has free access to the French website (https://mesdonnees.unicancer.fr/) to see if their data is being used for a research project and can choose to refuse or allow the reuse of their data.

## Ethics statement

Data analysis was approved in February 2023, by the local Data Protection Officer on behalf of French regulatory authorities (Commission Nationale de l'Informatique et des Libertéés, CNIL) according to the General Data Protection Regulation (GDPR). As explain in the following GDPR paragraph, all patients were informed of the possibility of reusing their health data for research purposes and did not express their opposition to this type of project.

## Results

In parallel with the daily use of digital health data and the development or integration of new computer software, we defined a data strategy making it possible to have reliable, accessible and easily reusable data for health data research aligned with international references (FAIR guiding principles) [17]. Our EMR mining process involves stages like obtaining data access, using SQL and tools for data extraction, and then refining the data, which includes both manual verification and automated processing via NLP. With CONSORE we have implemented an ETL (Extract, Transform, Load) pipeline that feeds into a structured data lake. Our next steps focus on enhancing NLP capabilities, collaborating with clinical experts, and expanding data sources (incorporing imaging data and genomic information) for a holistic patient view. This strategy is based on 4 main axes: data collection, data quality control, data analyses and data sharing (S2 Fig). Our EMR mining Process and our next steps are described in S3 Fig.

### Data collection strategies

For an effective data collection we first mined the use of EMRs, which provide a centralized repository for patient data and can be accessed by healthcare providers and researchers (pseudo-anonymously). CLB can collect data through patient-reported outcome measures (PROMs) via the portal patient myCLB, which provide valuable information about the patient experience and treatment outcomes. There are thus several ways of structuring the information of the EMR, either manually by the health professionals, or manually by the patient (essentially for adverse effects) or with the support of NLP. We have decided to primarily focus our efforts on a limited but comprehensive set of tumor-centered data.

### Patients

364 059 patients including 213 168 women (58.5%), 210 144 cancer patients (57.7%) and 39 331 (10.8%) patients with benign tumors. 135 446 are reported as dead (37.2%) with an updated vital status in EMRs up to date thanks to an automatic link with the CEPIDC platform which collects all the deaths of the French population [18]. This computerized update is essential for analyzing overall survival in retrospective studies and enabled to retrieve more than 78 365 death dates (Patients lost to follow-up and/or deceased outside our hospital). Among these cancer patients, 86 584 are in metastatic setting (41.2%) and 13 907 have 2 cancers at least (6.6%). Cancer patients' characteristics are summarized in Table 1. These raw data from the entire corpus of patients as well as the following graphs aim to demonstrate the technical feasibility of easily obtaining data but in no way to carry out a statistical evaluation of a neoplastic situation or a therapeutic strategy. Fig 2 represents the distribution of cancer diagnosis dates by year and Fig 3 represents the distribution of the patient's age at the time of cancer diagnosis. Fig 4 represents the proportion of survival of novo metastatic cancer patients (any cancer location and patients characteristics) according to the date of cancer diagnosis. In addition to these patient characteristics, the Léon Bérard center also has 1.3 million imaging exams (CT scans, MRIs, mammography, etc.), over 3 million biological tests, and 40 689 inclusions data from clinical trials.

*Side effects*. Since the beginning of the year 2022, we have implemented software (internally developed) that allows us to collect toxicity data and integrate it directly into medical consultation or hospital stay reports. These data are collected manually by medical doctors, coordinating nurses/ Advanced practice nurse or clinical research associates. After 1 year of use, we were able to collect 6394 adverse effects for 1248 cancer patients, of which 6188 were related to treatment and 437 had a CTCAE (Common Terminology Criteria for Adverse Events)

**Table 1. Characteristics of cancer patients.**

| | n | % |
|---|---|---|
| **Cancer Patients** | 210 144 | 100,0% |
| **Sex** | | |
| Women | 120 607 | 57,4% |
| Men | 89 537 | 42,6% |
| **Vital status** | | |
| Death | 93 754 | 44,6% |
| Alive | 116 390 | 55,4% |
| **Cancer type** | | |
| At least 2 primary cancers | 13 907 | 6,6% |
| Solid cancer | 191 734 | 91,2% |
| Hematologic cancer | 18 410 | 8,8% |
| metastatic disease (exclusively solid cancers) | 86 584 | 41,2% |
| De novo metastatic disease (exclusively solid cancers) | 32 334 | 15,4% |
| **Cancer location** | | |
| Head and Neck cancer | 7 700 | 3,7% |
| Breast cancer | 56 758 | 27,0% |
| Ovarian cancer | 5 882 | 2,8% |
| Lung cancer | 17 580 | 8,4% |
| Renal cancer | 5 199 | 2,5% |
| Prostate Cancer | 12 530 | 6,0% |
| Testicular cancer | 3 422 | 1,6% |
| Colo-rectal cancer | 10 638 | 5,1% |
| Oeso-gastric cancer | 5 930 | 2,8% |
| Pancreatic cancer | 3 136 | 1,5% |
| Melanoma | 9 477 | 4,5% |
| Bone sarcoma | 11 005 | 5,2% |
| Cancer unkown primary | 362 | 0,2% |
| Others | 42 115 | 20,0% |
| **Cancer laterality** | | |
| Right | 45 969 | 21,9% |
| Left | 45 727 | 21,8% |
| Bilateral | 1 093 | 0,5% |
| Unspecified/Not applicable | 98 945 | 47,1% |
| **Cancer treatment** | | |
| Surgery | 95 786 | 45,6% |
| Radiotherapy | 69 273 | 33,0% |
| Systemic treatment | 84 995 | 40,4% |
| Autologous Stem Cell Transplant | 1 108 | 0,5% |
| interventional radiology | 13 471 | 6,4% |

grade ≥ 2. These toxicities data are associated with symptoms reported by our cancer patients based on PROMs (7949 symptom forms completed by outpatients).

*Molecular markers.* Natural language processing can be used to extract insights from unstructured data, such as medical notes and patient report. NLP is a powerful tool for data structuration. NLP algorithms can be used to extract and categorize information from unstructured sources, such as EMRs. By using NLP, we can gain insights into patient outcomes, treatment patterns, and other relevant data that would otherwise be difficult or time-consuming to

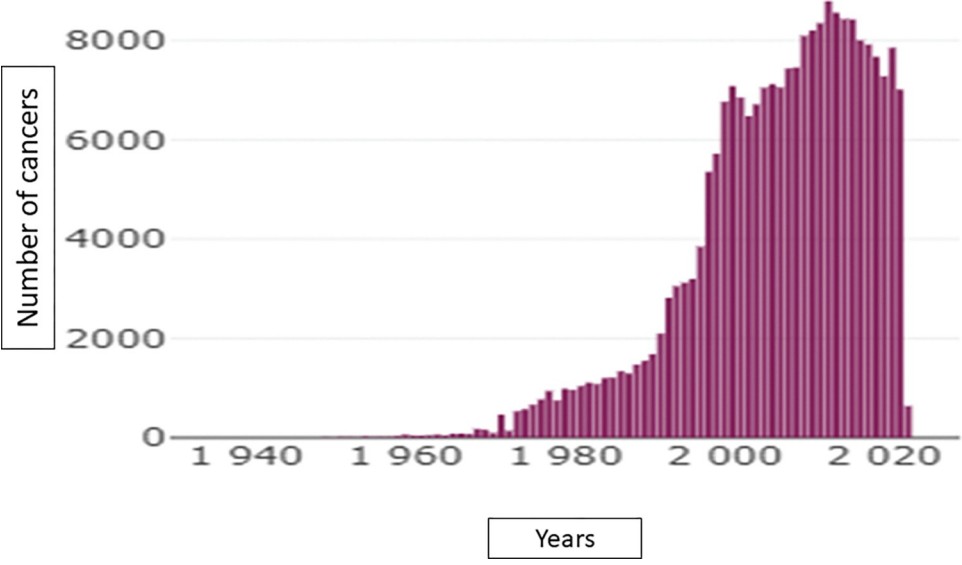

**Fig 2. Cancer diagnosis dates by year.**

extract from unstructured sources. To date, we have focused only on 4 theranostic biomarkers: estrogen receptor, progesterone receptor, Human epidermal growth factor receptor 2 (HER 2) by immunochemistry and fluorescence in situ hybridization and CPS (combined positive

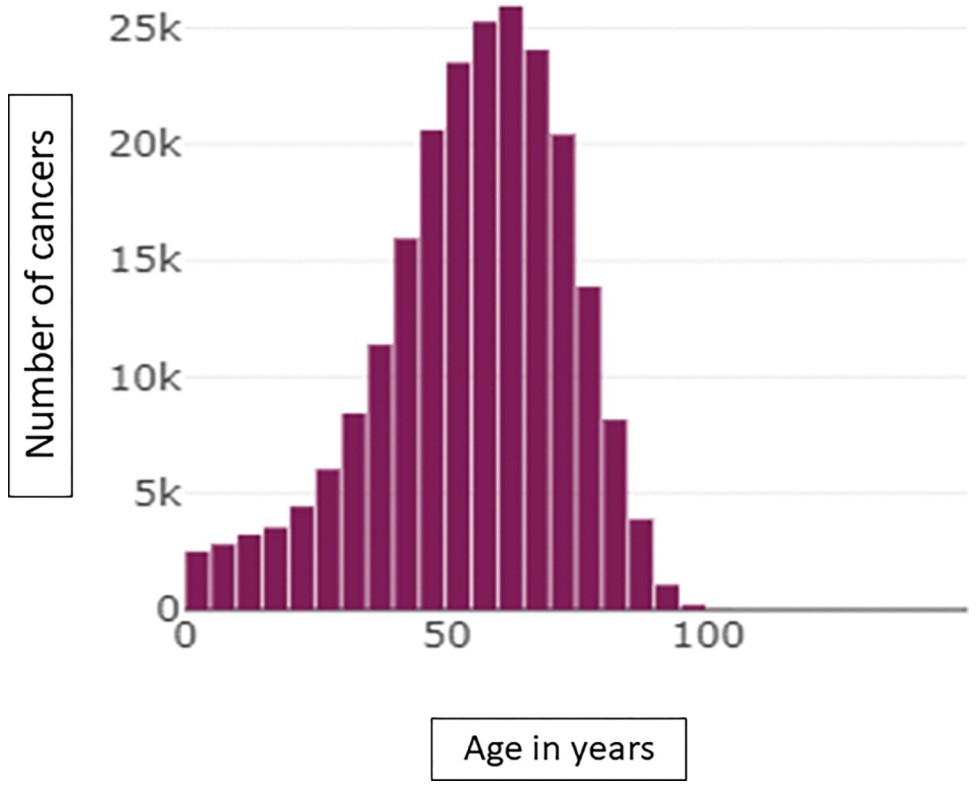

**Fig 3. Patient's age at the time of cancer diagnosis.**

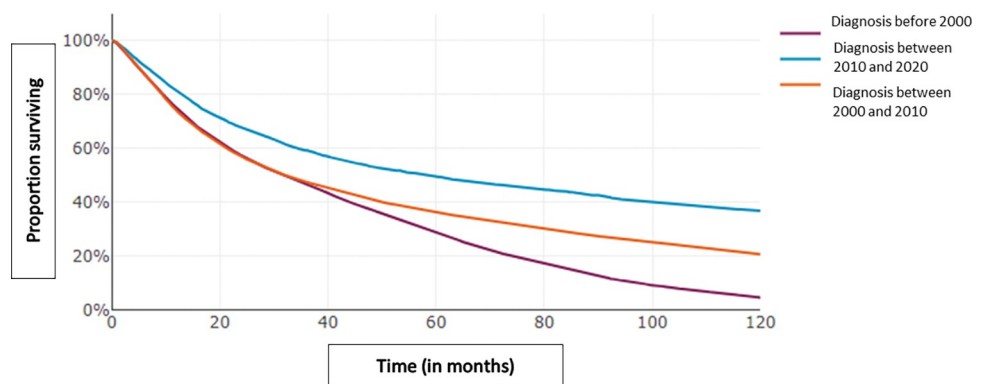

**Fig 4. Proportion of survival of novo metastatic cancer patients according to the date of cancer diagnosis.**

score). Our analysis covers 98% of the pathology results from the Léon Bérard center and allows us to obtain 95% of accuracy compared to a manually created breast cancer database. However, it remains to assess the accuracy of the results of these 4 tissue biomarkers on analyzes from other hospitals and also to expand the panel of biomarkers studied. One of the limitations of using NLP (Natural Language Processing) is the need for a sufficiently large and diverse training dataset to achieve adequate model performance. For some data sources, this requires significant human effort in structuring and annotating the data.

## Data quality control

Once data has been collected, it is essential to analyze it to extract meaningful insights and inform clinical decision-making. Quality control is another major point concerning data control process. Data quality can be defined as the absence of errors that matter and can be achieved with robust and documented processes. For example, we compared cancer location, cancer diagnosis dates and overall survival obtained by CONSORE with a validated monitored data issued from the PROFILER clinical trial (NCT01774409) whose data was manually integrated by specialized clinical research associates including 2025 metastatic cancer patients [19]. The primary goal of PROFILER trial was to analyze molecular profiles of advanced cancer patients and to recommend therapies based on genetic mutations. Thus, we can state that 98% of cancer localizations and 96% of morphologies are accurate and aligned with the International Classification of Diseases for Oncology, 3rd Edition (ICD-O-3) [20]. Quality control, currently done twice a year by CONSORE during version upgrades, focuses on data for research projects, with plans to establish a structured process to track authorship and validation dates for each tumor file.

## Data analysis strategies

Real-world data (RWD) studies are becoming increasingly important in the field of cancer research [21]. RWD refers to data that is collected from patients in the everyday setting of their medical care, outside of the controlled environment of clinical trials. RWD provides a more complete and accurate picture of patient outcomes and treatment patterns in the real world, as opposed to data collected in clinical trials, which may not fully reflect the complexity of real-world patient populations and treatment regimens [22]. RWD can also inform the development of new treatment strategies and help to identify disparities and unmet needs in cancer care. Additionally, RWD can support the development of predictive models and decision-making tools that can be used to optimize treatment and improve patient outcomes [23]. By

incorporating RWD into our data collection, analysis, and sharing strategies, we can take a more comprehensive and patient-centered approach to cancer research. This can lead to a better understanding of the disease and prognostic factors, improved patient outcomes, and the development of more effective treatments for cancer. For example, this organization has recently enabled us to publish quickly several articles on unknown or poorly studied pathologies such as COVID-19 [24,25] and the impact of vaccination, or second primary cancers [26,27].

## Data sharing strategies

Sharing data is critical for advancing cancer research and improving patient outcomes. Comprehensive cancer centers can share data with other institutions and researchers through data sharing agreements and consortia. For example, the French Health data hub is a platform that enables researchers to share data and collaborate on cancer research initiatives [28]. Many multicenter research projects are underway with this national platform [29–31]. Sharing data also enables the development of predictive models and enables researchers to test and validate new hypotheses. Additionally, data sharing agreements can help to ensure that patient privacy is protected while still enabling researchers to access the information they need. By sharing data, comprehensive cancer centers can work together to advance our understanding of cancer and improve patient outcomes. For instance, the implemented organization has enabled us to participate in research projects involving various hospitals and utilizing new artificial intelligence technologies [32,33].

## Data interoperability and the FHIR and OMOP standards

Data interoperability is a critical aspect of comprehensive cancer center data collection, analysis, and sharing strategies. Interoperability refers to the ability of different systems and applications to exchange and use data effectively. In the context of cancer research, interoperability is crucial for enabling the sharing of data between institutions and researchers and for facilitating the integration of data from multiple sources. To achieve interoperability, comprehensive cancer centers can adopt standards such as Fast Healthcare Interoperability Resources (FHIR) [34] and Observational Medical Outcomes Partnership (OMOP) [35]. OMOP is a data model specifically designed for observational medical research and provides a standardized framework for collecting, storing, and analyzing observational data. FHIR is a standard for exchanging healthcare data, including patient information and treatment outcomes, in a standardized format. By adopting these standards, comprehensive cancer centers can ensure that their data is consistent and compatible with other institutions and researchers, enabling the effective exchange and integration of data. This can improve the quality and relevance of cancer research and inform the development of new and more effective treatments. Given the work already undertaken on the OSIRIS data model, our ultimate goal is to align this model with OMOP and FHIR to facilitate data sharing during international projects.

## Discussion

We present here the development of a data strategy for health data research at the Léon Bérard cancer center with results of extraction as examples. The strategy is based on three main axes: data collection, data analysis, and data sharing. The CLB uses EMRs and PROMs to collect data on cancer patients, including tumor-centered data, vital status and patient characteristics. The CLB also collects toxicity data through software that integrates it directly into medical consultation or hospital stay reports. NLP algorithms are used to extract insights from unstructured data, such as medical notes, patient reports and more specifically histological reports.

CLB uses quality control to ensure the absence of errors in the collected data and believes that RWD studies provide a more complete and accurate picture of patient outcomes and treatment patterns than data collected in clinical trials.

The strategies for collecting, analyzing, and sharing data are essential for cancer centers [36,37]. Accurate and relevant data collection is the foundation of any effective data strategy, and cancer centers can use electronic medical records and patient-reported outcome measures to gather information about patients and their treatments [38]. Once data is collected, it is important to analyze it to extract actionable insights, using analysis techniques such as machine learning and natural language processing. Finally, data sharing is essential to advance cancer research and improve patient outcomes. Cancer centers can share data with other institutions and researchers through data sharing agreements and consortia [39].

In the European context, data collection, analysis, and sharing strategies are governed by the General Data Protection Regulation (GDPR), which establishes a framework for protecting the privacy and security of personal data, including health data [40]. Cancer centers must ensure that their data collection, analysis, and sharing strategies comply with GDPR and protect patient privacy [41].

Data interoperability is a critical aspect of data collection, analysis, and sharing strategies. It allows different systems and applications to exchange and use data effectively. To achieve interoperability, cancer centers can adopt standards such as FHIR and OMOP [42]. Adopting these standards enables cancer centers to ensure that their data is compatible with that of other institutions and researchers, facilitating the exchange and integration of data and improving the quality and relevance of cancer research.

Despite all these efforts, many elements still need to be worked on. For example, in the era of precision medicine, it is necessary to improve the completeness of our histological data and somatic and constitutional molecular biology data, which are currently very limited. The integration of NGS results with clinical data offers a more comprehensive understanding, allowing for a personalized treatment approach. Similarly, the pool of blood biology data is almost unusable, and work to structure the results and align variables with the international LOINC coding is mandatory. Additionally, digitization of pathology department is necessary so that whole slide images (WSI) become available for research projects and for the future use of artificial intelligence in routine. While Whole Slide Imaging (WSI) through digitization can provide an immense wealth of morphological details, its true potential is amplified when combined with other diagnostic methods. Thus, making WSI available is a step towards a holistic patient profile, but not an end in itself. Finally, many interoperability works, including aligning data models with OMOP/FHIR, are still necessary to facilitate international research projects.

In summary, data collection, quality control, analysis, and sharing strategies are essential to improving patient outcomes and advancing cancer research. Cancer centers must ensure that their strategies comply with current standards and regulations, while adopting innovative practices to ensure the efficiency and quality of data collection and analysis.

## Conclusion

Data is a critical asset for comprehensive cancer centers, and effective data strategies are essential for delivering high-quality care, advancing research, and improving patient outcomes. The adoption of electronic medical records, patient-reported outcome measures, and data sharing agreements are all key components of a successful data strategy. By implementing these strategies, comprehensive cancer centers can leverage data to drive advancements in cancer research and provide the best possible care to patients. In short, effective data strategies are essential to fighting cancer.

## Supporting information

**S1 Fig. Organizational chart.**
(TIF)

**S2 Fig. Data strategy schema.**
(TIF)

**S3 Fig. Electronic Medical Record mining process.**
(TIF)

## Author Contributions

**Conceptualization:** Pierre Heudel, Hugo Crochet, Thierry Durand, Jean-Yves Blay.

**Data curation:** Pierre Heudel, Hugo Crochet.

**Formal analysis:** Pierre Heudel, Hugo Crochet, Philippe Zrounba, Jean-Yves Blay.

**Methodology:** Pierre Heudel, Hugo Crochet, Thierry Durand.

**Project administration:** Hugo Crochet.

**Resources:** Hugo Crochet.

**Supervision:** Pierre Heudel, Hugo Crochet, Thierry Durand, Philippe Zrounba, Jean-Yves Blay.

**Validation:** Pierre Heudel, Hugo Crochet, Thierry Durand, Philippe Zrounba, Jean-Yves Blay.

**Visualization:** Pierre Heudel, Hugo Crochet.

**Writing – original draft:** Pierre Heudel, Hugo Crochet, Thierry Durand, Philippe Zrounba, Jean-Yves Blay.

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
