## [Decision Letter · Decision Letter 0]

1 Aug 2023

PDIG-D-23-00112

From Data Strategy to Implementation to advance cancer research and cancer care: A French comprehensive cancer center Experience

PLOS Digital Health

Dear Dr. HEUDEL,

Thank you for submitting your manuscript to PLOS Digital Health. After careful consideration, we feel that it has merit but does not fully meet PLOS Digital Health's publication criteria as it currently stands. Therefore, we invite you to submit a revised version of the manuscript that addresses the points raised during the review process.

Please submit your revised manuscript within 60 days Sep 30 2023 11:59PM. If you will need more time than this to complete your revisions, please reply to this message or contact the journal office at digitalhealth@plos.org. Please include the following items when submitting your revised manuscript:

We look forward to receiving your revised manuscript.

Kind regards,

Ludwig Christian Giuseppe Hinske, M.D.

Academic Editor

PLOS Digital Health

Journal Requirements:

1. Please ensure that Funding Information and Financial Disclosure Statement are matched.

2. In the Funding Information you indicated that no funding was received. Please revise the Funding Information field to reflect funding received.

3. Please provide separate figure files in .tif or .eps format only and remove any figures embedded in your manuscript file. Please also ensure that all files are under our size limit of 10MB.

4. Tables should not be uploaded as individual files. Please remove these files and include the Tables in your manuscript file as editable, cell-based objects. For more information about how to format tables, see our guidelines:

https://journals.plos.org/digitalhealth/s/tables

5. We do not publish any copyright or trademark symbols that usually accompany proprietary names, eg ©, ®, ™ (e.g. next to drug or reagent names). Please remove all instances of trademark/copyright symbols throughout the text, including ® on page 10.

Additional Editor Comments (if provided):

Reviewers' comments:

Reviewer's Responses to Questions

**Comments to the Author**

1. Does this manuscript meet PLOS Digital Health’s publication criteria? Is the manuscript technically sound, and do the data support the conclusions? The manuscript must describe methodologically and ethically rigorous research with conclusions that are appropriately drawn based on the data presented.

Reviewer #1: Partly

Reviewer #2: Yes

2. Has the statistical analysis been performed appropriately and rigorously?

Reviewer #1: N/A

Reviewer #2: N/A

3. Have the authors made all data underlying the findings in their manuscript fully available (please refer to the Data Availability Statement at the start of the manuscript PDF file)?

Reviewer #1: No

Reviewer #2: No

4. Is the manuscript presented in an intelligible fashion and written in standard English?

Reviewer #1: No

Reviewer #2: Yes

5. Review Comments to the Author

Reviewer #1: Dear authors,

let me first take the opportunity to thank you for contribution to a topic in a rapidly evolving field that urgently needs to be streamlined regarding standardized strategies for processing data.

Research published in PLOS Digital Health should fulfill each of the following criteria:

1. Originality

2. High importance and broad interest to the community of researchers, engineers and clinicians working in the field of digital health

3. High methodological rigor and ethical standards

4. Substantial evidence for its conclusions

5. Clearly outlined utility and accessibility for the broader community

6. Follow appropriate standards and practice of open science

I therefore want to outline, that to fulfill all criteria, especially to add value to the scientific community, your manuscript needs to be supplemented and revised from my point of view. A major issue is that the manuscript is too superficial in some segments. I sometimes find it hard to recognize what qualifies your currently described work as original research as it is a description of your strategy, enriched by very few preliminary data gained from your specific pipeline at CLB. Nevertheless, I believe that with major revisions it should be able to improve this manuscript so it will be adding value to the scientific community.

In your Introduction you criticize the rather too theoretical approach of scientific cancer societies and emphasize your practical approach with concrete examples. However, on the main elements your manuscript must face the same reproach.

In Material & Methods you summarize the historical development of the EMG at CLB. I suggest you add a graphical overview of the development over time to summarize your text. From such an overview it would become clear that the pace of development is rapidly increasing. Try to be as informative as possible when describing EMR elements regarding data structure, interoperability, and data sources as well as data providers. In addition, this figure would be of benefit for describing your vision of an idealized data lake.

Moreover, a second figure to visualize the information system directorate and their respective interfaces would be helpful. Both figures together allow for a concrete glimpse of work in progress at CLB and can thus be of guidance for other centers that are struggling with the same task.

As for the results section, please prepare a figure for your 4-axis strategy that includes a preview of the results and gives the precise measures, your data strategy includes. Furthermore, it is unclear to which extent the EMR can be mined so far, as you do not precisely describe the process of mining, extracting and data wrangling. How much of these steps can be attributed to manual or automated steps, e.g. with NLP? Is there an ETL pipeline or a data lake? It would be helpful to know to which extent pipelines are automized so far and what your next steps are to further develop your data strategy.

For the patients-section, can you state if the characteristics summarized in Table 1 are raw extracted data or if they have been modified. First, the current table is only partially suited to gather further insights for a researcher: For the cancer type, metastatic, de novo metastatic and localized disease should be assigned to solid cancer as subsections. Moreover, cancer location is an incorrect label as you are describing the respective cancer entity. Cancer laterality is only relevant to distinct entities, such as colorectal or breast cancer but not for the overall subset. Cancer treatments as a summary is interesting but would be even more insightful, if therapeutic modalities were assigned to each cancer entity. The summary should focus on the three main pillars: systemic, surgical and radiotherapy. With regard to future trends, you should think about adding a fourth pillar, i.e. cellular therapies as well. These include CAR-T-cell therapy, bispecific antibodies and allogenous stem cell transplants. As a next you could aim to have this arranged over time.

In conclusion, I suggest that you define a meaningful summary table with the oncology team and try to arrange your ETL pipeline accordingly. If, what you have summarized in the table is the raw output, it should be arranged differently, as the way the data is presented now, it is not informative.

For the acquisition of side effects, you describe to have implemented a software. Please be specific and describe the way the software works and if it is a commercial software or self-made and if or how it was implemented into your pipeline.

For the data qualification section, you compared data extracted via CONSORE with the validated data from the PROFILER trial. Please be more specific, describe the trial in a few sentences.

What you describe in data analysis strategies, data sharing strategies, general data protection regulation (GDPR) and data interoperability and the FHIR and OMOP standards is not part of the results section but rather reads like part of the material & methods section.

In your discussion, you correctly state the current sparsity of your data but in my view, you lack a future vision of how your data strategy can be developed further in a stepwise fashion to comprehensively characterize cancer patients at CLB. Is it necessary to digitize to make WSI available? Or is it of more value to have the results of molecular diagnostics such as NGS at hand to combine them with clinical data to get meaningful information? Finally, I was confused to read about three main axes in the discussion as you have mentioned four main axes at the beginning of the results section.

To conclude, there a several mistakes in grammar in spelling which must be corrected. Please find enclosed a selection:

Line 66): Cancer is …

Line 71/72): … a variety of …

Line 114): The project uses …

Line 66): Cancer is …

Reviewer #2: The authors present on an approach to data collection,

analysis, and sharing to enhance their understanding of cancer and improve patient outcomes.

Leveraging advanced data analytics techniques and developing effective data sharing policies can help

cancer centers effectively harness the power of data to drive progress in cancer research.

Major: 

1. Section Data Qualification: Is the quality control a procedure only performed once? Or are there workflows in place or plans to conduct routinely data quality assessments and reports? Are there standards used to follow such routine workflows?

2. Data Analysis strategies: Please provide references for you statements on read world data studies in this paragraph(page 12).

3. Section Data sharing strategies: Please add results to this section. Maybe you can explain (diagram?) your workflows in how you were able to successfully share your data with outside institutions?

4. Section GPDR: Are the mentioned two levels of patient information available? Can they be linked or referenced here? Was there feedback from patients or researchers on the usability of this information?

5. Section Interoperability: Are the referenced papers results of the project? If not, can you focus this section on your results or describe the planned results of the project abit more in depth?

Minor:

6. Please check if section „Data Qualification“ should be renamed to „Data quality control“

7. Please check if capitalization of letters in title and sections are compliant with the journal guideliens

6. PLOS authors have the option to publish the peer review history of their article (what does this mean?). If published, this will include your full peer review and any attached files.

**Do you want your identity to be public for this peer review?** For information about this choice, including consent withdrawal, please see our Privacy Policy.

Reviewer #1: No

Reviewer #2: No

---

## [Decision Letter · Decision Letter 1]

17 Oct 2023

PDIG-D-23-00112R1

From Data Strategy to Implementation to advance cancer research and cancer care: A French comprehensive cancer center Experience

PLOS Digital Health

Dear Dr. HEUDEL,

Thank you for submitting your revised manuscript to PLOS Digital Health. As you can see, the reviewers only had minor suggestions.

Please submit your revised manuscript within 30 days Nov 16 2023 11:59PM. If you will need more time than this to complete your revisions, please reply to this message or contact the journal office at digitalhealth@plos.org. Please include the following items when submitting your revised manuscript:

We look forward to receiving your revised manuscript.

Kind regards,

Ludwig Christian Giuseppe Hinske, M.D.

Academic Editor

PLOS Digital Health

Journal Requirements:

Additional Editor Comments (if provided):

Reviewers' comments:

Reviewer's Responses to Questions

**Comments to the Author**

1. If the authors have adequately addressed your comments raised in a previous round of review and you feel that this manuscript is now acceptable for publication, you may indicate that here to bypass the “Comments to the Author” section, enter your conflict of interest statement in the “Confidential to Editor” section, and submit your "Accept" recommendation.

Reviewer #1: (No Response)

Reviewer #2: All comments have been addressed

2. Does this manuscript meet PLOS Digital Health’s publication criteria? Is the manuscript technically sound, and do the data support the conclusions? The manuscript must describe methodologically and ethically rigorous research with conclusions that are appropriately drawn based on the data presented.

Reviewer #1: Partly

Reviewer #2: Yes

3. Has the statistical analysis been performed appropriately and rigorously?

Reviewer #1: N/A

Reviewer #2: N/A

4. Have the authors made all data underlying the findings in their manuscript fully available (please refer to the Data Availability Statement at the start of the manuscript PDF file)?

Reviewer #1: Yes

Reviewer #2: Yes

5. Is the manuscript presented in an intelligible fashion and written in standard English?

Reviewer #1: Yes

Reviewer #2: Yes

6. Review Comments to the Author

Reviewer #1: Dear Editor and Authors,

thank you for having me review the revisions for the manuscript entitled "From Data Strategy to Implementation to advance cancer research and cancer care: A French comprehensive cancer center Experience".

I will briefly comment on my points previously raised:

First of all, I would like to ask the supplemental figures to be added to the manuscript, as I have otherwise no way to evaluate them.

1) This comment was indicated to ask for more detail in the manuscript. This request was complied with in part throughout the revisions.

2) Figure 1 has been composed and helps to get a quick overview of the many measures utilized. However it still lacks a good overview and more detail (What is a "Tumoral resources center"; what does a "Request for medical advice mean?" Is it a simple consultation request form or something more sophisticated?. A figure description is missing. Moreover, the colors chosen (pink) make it harder to read. I moreover suggest to add some kind of chronological order. The formatting is mixed within the figure. Finally I suggest changing the title "Applications patient centered" as I am struggling to see how it is related to the contents.

3) I would like to see Supplementary Fig 1

4) I would like to see Supplementary Fig 2 and I appreciate your detailed description of the EMR mining process. However, I am convinced the reader benefits from the details as well, which is why I suggest integrating them to the manuscript of within another figure

5) This comment has been sufficiently addressed.

6) This comment has been partly addressed. However, if the sole purpose is to demonstrate feasibility, reducing the number of figures and moving these additional figures to the supplement adds to the clarity and conciseness of the manuscript.

7) This comment has been sufficiently addressed.

8) This comment has been sufficiently addressed.

9) This comment has been sufficiently addressed.

10) This comment has been sufficiently addressed.

11) Please once more check for errors in grammar an spelling, as a well-written manuscript is easier to follow: "Cancer is a disease ..." etc.

Reviewer #2: (No Response)

7. PLOS authors have the option to publish the peer review history of their article (what does this mean?). If published, this will include your full peer review and any attached files.

**Do you want your identity to be public for this peer review?** For information about this choice, including consent withdrawal, please see our Privacy Policy. 

Reviewer #1: No

Reviewer #2: No

---

## [Editor Report · Decision Letter 2]

20 Nov 2023

From Data Strategy to Implementation to advance cancer research and cancer care: A French comprehensive cancer center Experience

PDIG-D-23-00112R2

Dear Dr HEUDEL,

We are pleased to inform you that your manuscript 'From Data Strategy to Implementation to advance cancer research and cancer care: A French comprehensive cancer center Experience' has been provisionally accepted for publication in PLOS Digital Health.

Best regards,

Ludwig Christian Giuseppe Hinske, M.D.

Academic Editor

PLOS Digital Health